Comparative analysis of SNP candidates in disparate milk yielding river buffaloes using targeted sequencing

Menon Ramesh
Patel Anand B.
Joshi Chaitanya cgjoshi@rediffmail.com
Department of Animal Biotechnology, Anand Agricultural University , Anand , India
Nakai Kenta
Electronic publication date: 2016 Jul 7
Publication date: 2016
Volume: 4
Electronic Location ID: e2147
Received 2016 Jan 30; Accepted 2016 May 27
Copyright: © 2016 Menon et al.
Copyright year: 2016
Copyright holder: Menon et al.
License: This is an open access article distributed under the terms of the Creative Commons Attribution License, which permits unrestricted use, distribution, reproduction and adaptation in any medium and for any purpose provided that it is properly attributed. For attribution, the original author(s), title, publication source (PeerJ) and either DOI or URL of the article must be cited.
License URL: https://creativecommons.org/licenses/by/4.0/

Keywords: Buffalo, Single nucleotide polymorphism, Milk-yield, Quantitative trait loci, Mammary transcriptome, Exome sequencing

Funding: This study was funded by Department of Biotechnology, Ministry of Science and Technology, Government of India. The funders had no role in study design, data collection and analysis, decision to publish, or preparation of the manuscript.

==============================
River buffalo (Bubalus bubalis) milk plays an important role in economy and nutritious diet in several developing countries. However, reliable milk-yield genomic markers and their functional insights remain unexposed. Here, we have used a target capture sequencing approach in three economically important buffalo breeds namely: Banni, Jafrabadi and Mehsani, belonging to either high or low milk-yield group. Blood samples were collected from the milk-yield/breed balanced group of 12 buffaloes, and whole exome sequencing was performed using Roche 454 GS-FLX Titanium sequencer. Using an innovative approach namely, MultiCom; we have identified high-quality SNPs specific for high and low-milk yield buffaloes. Almost 70% of the reported genes in QTL regions of milk-yield and milk-fat in cattle were present among the buffalo milk-yield gene candidates. Functional analysis highlighted transcriptional regulation category in the low milk-yield group, and several new pathways in the two groups. Further, the discovered SNP candidates may account for more than half of mammary transcriptome changes in high versus low-milk yielding cattle. Thus, starting from the design of a reliable strategy, we identified reliable genomic markers specific for high and low-milk yield buffalo breeds and addressed possible downstream effects.

Introduction

Dairy farming has been a vital part of agriculture for several thousands of years, and have a high impact in social, health and economic conditions of several Asian countries (http://www.fao.org). Notably, India is the largest producer of milk in the world, and buffalo’s milk account for about half of India’s milk production (http://www.fao.org). Milk or milk products form the primary source of fat and protein in an average Indian diet, and buffalo dairy farming is a life sustaining business for millions of small-scale dairy farmers in the country. Economically important Indian buffalo breeds include: Mehsani, Jafarabadi, Nili-Ravi, Surti, Murrah, Pandharpuri, Banni etc. Despite their multi-faceted significance in countries like India, a few effort has been made so far to improve buffalo breeding programs (Singh et al., 2009). It is well understood that milk yield in dairy animals have a major genetic component (Cole et al., 2009). In the recent years, genomic selection in cattle using molecular markers such as SNPs has been proved to be highly efficient (Berglund, 2008). A multi-breed analysis will facilitate precise dissection of milk-yield makers and their molecular framework (Goddard & Hayes, 2009; Melka & Schenkel, 2012; Olson, VanRaden & Tooker, 2012). Recently, taking advantage of next-generation sequencing with targeted DNA capture technologies are proved to be an efficient and cost-effective approach for high-throughput SNP discovery (Hirano et al., 2013).

Here, we designed a whole exome capture custom array that target exons and untranslated regions (UTRs) of the cattle genome to perform whole-exome/targeted sequencing and conducted exome sequencing to identify variants in three recognized milch breeds of river buffalo native to Gujarat state of India viz. Banni, Mehsani, and Jafrabadi breeds. This study design allows us to identify the genetic variants specific for buffalo breeds.

Material and Methods

Sample collection and genomic DNA extraction

We recruited 12 water buffaloes, which were biological replicates from three breeds namely Banni, Mehsani and Jafrabadi, belonging to either high or low-milk yield category from the Gujarat state of India (Table 1). The gDNA was isolated from blood samples using a Qiagen DNeasy Blood and Tissue kit (Qiagen Corp., CA, USA) and the resultant DNA was quantified using Qubit® dsDNA BR Assay (Invitrogen Corp., CA, USA) and integrity was confirmed by agarose gels. Blood drawings were conducted in accordance with regulations and prior approval by the institutional animal ethics committee of Anand Agricultural University, Gujarat, India.

Table 1 Summary of samples recruited for the analysis.

Group	Sample ID	Breed	Total milk yield (liters)	
Low milk yield	BLP1	Banni	1,680	
BLP2	1,400	
MLP2	Mehsani	1,401	
MLP5	1,449	
JLP1	Jafarabadi	571	
JLP2	583	
High milk yield	BHP1	Banni	6,440	
BHP2	5,880	
MHP1	Mehsani	4,104	
MHP2	4,091	
JHP1	Jafarabadi	3,186	
JHP2	2,947	

Probe design, target enrichment and exome sequencing

We obtained the intended targets (all coding exons, 3′UTR and 5′UTR exons) of cattle (Btau_4.6.1/bos Tau7) from RefGene tables of UCSC genome browser. Further, the custom probe design was performed by NimbleGen (Roche, Germany) which are compatible with Roche GS-FLX Titanium sequencer.

Rapid library for each sample was prepared from ∼1 μg of gDNA separately and multiplexed according to manufacturer’s protocol (Roche, Germany) using high quality DNA. Final libraries were used to set up hybridization reaction at isothermal temperature of 47 °C for 68–72 h in thermal cycler, with custom designed probes as per manufacturer’s protocol (NimbleGen). Captured DNA libraries were quantified spectrophotometrically, and evaluated electrophoretically with high sensitivity DNA assay on Agilent Bioanalyzer 2100 (Agilent). Finally, the libraries were sequenced on Roche 454 GS-FLX Titanium instrument according to manufacturer’s protocol (Roche, Germany). The raw sequence data will be deposited in a public repository.

Raw reads filtering, alignment and variant detection

Raw reads were filtered based on phred quality score ≥ 20 and length ≥ 50 bp using QTrim tool (Schmieder & Edwards, 2011). The resultant sequence reads were mapped against Bos taurus genome build 4.6.1 using ‘bwa-mem’ module of BWA v 0.7.5a (Li & Durbin, 2010). Potential PCR duplicates were removed using ‘MarkDuplicate’ module of Picard Tools. SNPs passing the criteria depth ≥ 5 and quality score ≥ 30 were identified using SAMtools, FreeBayes and VarScan tools, through pooled variant calling strategy (Li et al., 2009; Garrison & Marth, 2012; Koboldt et al., 2012). To deduce the shared (MultiCom) and specific SNPs from each tool, we used VCFtools package v0.1.11 and basic Linux commands (Danecek et al., 2011). The SNPs belonging to high-yield and low-yield were compared using VCFtools package v0.1.11 (Danecek et al., 2011). The specific SNPs (high or low yield) refers to the SNPs that were present in any of the sample in the given group which are not present or did not pass the SNP calling threshold in the other group.

Annotation and QTL dataset

The resulting sequence ontology (SO) and candidate gene annotation of SNPs specific for high and low yield groups was performed using SnpEff v 3.4 (Cingolani et al., 2012). The candidate gene enrichment analysis was performed in the GeneCodis tool and the enriched pathways and gene ontology (GO) terms were identified using Hypergeometric test followed by Benjamini-Hochberg’s correction (p-value < 0.01). Cattle QTL candidate genes belonging to milk-yield and milk-fat categories were obtained from the AnimalQTLdb online resource (Hu et al., 2013).

Mammary gland transcriptome analysis

From NCBI-GEO database, we obtained the whole gene expression data of 8 samples derived from mammary gland of Holstein Friesen cattle belonging to high or low milk yield (Accession: GSE33680). The gene expression was measured using Agilent-023647 v2 microarray, comprising of 45,220 oligo-nucleotide probes representing cattle genes. In the dataset, there were four samples in each for high-milk yield and low-milk yield groups. GEO2R implementation of LIMMA ebayes test was used to identify differentially regulated genes between high and low milk yield groups (p-value < 0.01).

Results

Samples and overview of sequence data

In order to identify SNPs specific for high and low milk-yield in Indian buffalo breeds, we designed two balanced sample groups of healthy Indian buffaloes with prominent difference in milk yields. Animals with fertility issues, inconsistent diets or other known disorders were excluded from the study. The custom designed capture probes based on UCSC RefTable data of Bos taurus genome version 4.6.1/bosTau7 covered 125,679 exons, 14,084 3′UTRs and 16,574 5′UTRs. The sequencing experiment of 12 samples generated about 2.49 gigabases encompassed in ∼6.5 million sequence reads with average length of 380 bases (Table 2). We obtained ∼68% average on-target capture efficiency across samples and approximately 98% of filtered reads were mapped to Bos taurus 4.6.1 reference sequence (NCBI), with the average per sample depth ≥ 5× (Table S1).

Table 2 Summary of sequence data.

Sample ID	Raw dataset	Filtered dataset	
Total reads	Base count (Mb)	Mean read length	No. of reads	Base count (Mb)	Mean read length	% passed	Target covered (%)	
BLP1	468,655	179.2	382	434,193	149.6	344	92.6	70.6	
BLP2	544,069	206.1	379	503,649	173	343	92.6	71.7	
BHP1	604,742	229.7	380	555,099	195	351	91.8	63.3	
BHP2	629,606	240.7	382	580,983	204.8	352	92.3	65.5	
MLP2	573,466	213	371	530,089	176.3	332	92.4	67.2	
MLP5	437,957	165.4	378	407,996	137.1	336	93.2	73.3	
MHP1	544,403	197.3	362	498,613	163.2	327	91.6	70.2	
MHP2	657,400	238.4	363	602,554	197.3	327	91.7	62.8	
JLP1	632,961	249	393	594,251	213.9	359	93.9	65.3	
JLP2	576,284	227.3	394	541,456	195.4	360	94	66.3	
JHP1	447,055	172.3	385	413,569	145.1	350	92.5	76.4	
JHP2	445,066	180	404	413,702	151.7	366	93	74.3	
	6,561,664 (total)	2,498 (total)	381 (average)	6,076,154 (total)	2,102 (total)	346 (average)	92.6 (average)	67.8 (average)	

Identification of SNPs specific for high and low milk-yield using MultiCom approach

We grouped the samples based on milk-yield (high and low milk-yield groups), and performed the SNP analysis separately for each group, using cattle as reference. Initially, we used three variant calling tools namely: SAMtools, FreeBayes and VarScan with the consistent thresholds of base depth ≥ 5 and quality score ≥ 30 for all the tools. In the high milk-yield group, SAMtools detected 1.097 million SNPs. FreeBayes and VarScan reported 1.101 million and 1.090 million SNPs respectively (Fig. 1A–1C). Even though high concordances were observed between the outcomes from the three tools, several thousands of SNPs (6.5–9.3% of output from each tool) were tool-specific or non-reproducible SNP calls (Fig. 1A). On the other hand, in the low milk-yield group, SAMtools identified 1.389 million SNPs, followed by 1.103 and 1.098 million SNPs respectively by VarScan and FreeBayes. In the low milk-yield group, the tool-specific SNP calls for SAMtools was the same as that of high-production dataset (6.5%), but a considerable reduction of tool–specific calls was observed for VarScan and FreeBayes (4.6 and 3.8%, respectively) outputs, even though the total outcome by these two tools were comparable with the high milk-yield group (Fig. 1A). We hypothesized that the non-reproducible SNPs were potential false positive calls. To test this, we compared the transition/transversion (ts/tv) ratios of SNPs which were tool-specific and shared, separately for the two groups. A highly consistent ts/tv range was observed for the shared SNPs, ranging from 2.61 to 2.65. On contrary, the tool-specific or non-reproducible SNPs showed a dynamic range of lower ts/tv ratios, ranging from 1.60 to 2.38 (Fig. 1D). Strikingly, the difference in the ts/tv ratios were highly significant (p-value < 0.005) (Fig. 1D). Next, using SO annotation by snpEff tool, we compared the regions in the reference genome in which the SNPs are located. Notably, it was observed that the inter-genic region SNPs were almost doubled in case of tool-specific SNP calls, consistently in three tools and the two groups (Fig. 1E). Therefore, we considered only those SNPs which were reproduced by at least two tools (hereon called MultiCom approach). There were a total of 1.203 million SNPs in high milk-yield group and 1.315 million SNPs for low milk-yield group (Fig. 1A–1C). Interestingly, in almost all cases, SNP selection through MultiCom approach facilitated an increased discovery rate (up to 20%) compared to the outcome from each tool. Finally, we compared MultiCom outcomes in high milk-yield and low milk-yield groups, and have found that 255,741 SNPs (21.3%) were specific for high yield group, and 367,674 SNPs (28.0%) were specific for low yield group. On the other hand, the 947,646 SNPs were common to the two groups which may be generic SNPs in buffalo, as the reference genome was cattle.

Figure 1 SNPs discovery using MultiCom approach.

(A-B) shows the MultiCom SNP discovery in the high and low production respectively. The total number of high quality SNPs detected by SAMtools, VarScan and Freebayes tools are given. In (A and B) the numbers given in uncoloured region inside the Venn diagram is the number of non-reproducible or tool-specific SNPs. MultiCom approach identified 1.203 million and 1.315 million SNPs respectively in high and low production group. (C) shows the number of shared and specific SNPs in high and low production groups. (D) Comparison of ts/tv ratios in tool-specific and shared SNPs outputted by each of the three tools in high and low production groups. The difference in ts/tv ratio is statistically significant (p-value < 0.005) (E) Percentage of intergenic SNPs detected by each of the three tools in high and low production groups.

Candidate gene prediction and comparison with milk-yield and milk-fat QTL dataset

Using snpEff prediction candidate genes were determined for high and low milk-yield specific SNPs (Table S2). The snpEff annotation mapped the 255,741 high yield specific SNPs in 7,212 genes, and the 367,674 low yield specific SNPs in 8,284 coding genes. Even though the SNPs were specific for the two groups, we have found that the 5,037 genes were common to high and low milk-yield groups. On the other hand, there were 2,175 genes specific for high yield and 3,247 for low milk-yield. In order to determine whether these genes were present among the reported milk-yield and milk-fat related genes in QTL region, we obtained the list of candidate genes in milk-yield and milk-fat categories of cattle resources of the animal genome data repository. Remarkably, of 74 milk-yield candidates, 51 genes were present in our results, among which 26 were common to high and low milk-yield group and 17 genes specific for low yield specific genes, and 8 were high specific genes (Fig. 2A). On the other hand, of 91 milk-fat candidates, 30 genes were common to high and low-milk yield group, 24 and 15 candidate genes were specific for low and high milk-yield groups respectively (Fig. 2B). On the whole, about 70% of the QTL candidate genes in respective categories belonging to cattle genome annotation were present among the genes the genes identified by our study in buffalo breeds.

Figure 2 Gene level comparison of SNPs specific in high and low milk-yield groups with QTL dataset.

(A) Overlapping genes in milk-yield category. The first column represents high and low milk yield specific SNPs located in 26 common genes. The second and third column respectively indicates the 17 low-milk yield specific and 8 high-milk yield specific candidate genes. (B) Overlapping genes in milk-fat category. The first column represents high and low milk yield specific SNPs located in 30 common candidate genes. The second and third column respectively indicates the 24 low-milk yield specific and 15 high-milk yield specific candidate genes.

Enrichment analysis of yield specific candidate genes

Next, we performed the GO enrichment analysis of the candidate genes, specific for high and low milk-yield groups with the threshold of FDR corrected p-value < 0.01. In the GO molecular functions category, eight GO terms were enriched in high milk-yield group. On the other hand, 21 terms were enriched in low-milk yield group, among which four terms were common to GO terms belonging to high milk-yield group (Fig. 3). The common terms were metal ion binding, nucleotide binding, zinc ion binding and cytokine activity. Interestingly, several GO terms in the low milk-yield category which were related to transcriptional regulation, such as sequence-specific DNA binding transcription factor activity (GO:0003700), sequence-specific DNA binding (GO:0043565) and transcription regulatory region DNA binding (GO:0044212) (Fig. 3A). Collectively, 140 genes were found to be contributing to the statistical enrichment of these three GO terms, which were specific for low-milk yield group (Fig. 3C).

Figure 3 Gene ontology enrichment analysis of high and low specific gene candidates.

(A) Twenty one gene ontology (molecular functions) terms enriched in low-yield specific group. The number of hits associated with each term along with the FDR corrected p-value is given. (B) Eight gene ontology (molecular functions) terms enriched in high-yield specific group. (C) The transcriptional regulators (140 genes) highlighted by the GO analysis in low-yield group.

In order to obtain the clues about the functional consequences of the candidate genes specific for high and low-yield specific groups, we performed the KEGG pathway enrichment analysis using Genecodis tool. There were 15 pathways enriched for the high yield specific group and 32 pathways for the low-yield specific group (Table S3). However, we have noticed that several of the pathways were related to human diseases. Further, after excluding human disease specific pathways, there were 8 pathways present in high-yield group and 19 pathways for the low milk yield group (Table 3). The most significant pathways in the low milk-yield group were Oxidative phosphorylation, Toll-like receptor signalling, cytokine-cytokine receptor interaction etc. On the other hand, the most significant among the high-yield group were Jak-STAT signalling pathway, Wnt signalling pathway, ErbB signalling pathway etc. Surprisingly, Toll-like receptor signalling and MAPK signalling pathway were enriched in both high and low milk-yield groups by distinct set of genes falling in the same pathway (Figs. S1–S3).

Table 3 Pathways enriched in low and high milk-yield groups.

KEGG pathways	Group	Genes	FDR p-value	
Oxidative phosphorylation	L	33	1.53E-07	
Toll-like receptor signalling pathway*	L	20	3.39E-05	
	H	13	4.54E-03	
Cytokine-cytokine receptor interaction	L	26	1.08E-04	
Jak-STAT signalling pathway	H	18	1.96E-04	
Melanogenesis	L	14	6.75E-04	
Spliceosome	L	21	9.81E-04	
Viral myocarditis	L	11	1.03E-03	
Wnt signaling pathway	H	17	1.91E-03	
RIG-I-like receptor signaling pathway	L	12	2.99E-03	
Osteoclast differentiation	L	17	3.25E-03	
Toxoplasmosis	L	17	3.25E-03	
Adipocytokine signaling pathway	L	12	3.40E-03	
Protein digestion and absorption	L	10	3.91E-03	
ErbB signaling pathway	H	10	4.14E-03	
Natural killer cell mediated cytotoxicity	H	13	4.36E-03	
Leukocyte transendothelial migration	H	14	4.43E-03	
Cell cycle	L	17	4.87E-03	
MAPK signaling pathway*	L	27	5.01E-03	
	H	21	5.28E-03	
Vasopressin-regulated water reabsorption	L	9	6.02E-03	
VEGF signaling pathway	H	10	6.79E-03	
Collecting duct acid secretion	L	7	7.38E-03	
Axon guidance	L	15	7.50E-03	
Apoptosis	L	13	7.56E-03	
Cell adhesion molecules (CAMs)	L	15	8.50E-03	
Protein processing in endoplasmic reticulum	L	20	9.14E-03	
Notes:

L, Low milk-yield; H, High milk-yield.

* Common pathways in L and H groups.

Comparison of buffalo milk-yield specific candidate genes with milk-yield DEG (Differentially Expressed Genes) in cattle

Next, we hypothesized that the SNPs in the candidate gene may exert changes in the gene expression that could have an effect on milk yield. In order to verify this, we obtained the mammary cell transcriptome dataset belonging to high and low milk-yielding cattle (see Materials and Methods). LIMMA analysis between the high milk-yield and low milk-yield group detected 1,056 annotated differentially expressed genes, in which 579 gene were up-regulated and 477 genes were down-regulated among the low milk-yield samples (Fig. 4; Table S4). Quite interestingly, it was found that 582 out of 1,056 DEG were buffalo SNP candidate genes in low yield specific, high yield specific or common to both groups (Fig. 4). Among these 582 overlapping DEG, 196 genes (118 up-regulated and 78 down-regulated) belonged to low yield specific group, 139 genes (68 up-regulated and 71 down regulated) were high yield specific group. On the other hand, 247 DEG (147 up-regulated and 100 down-regulated in low milk yield) were common to high and low milk-yield groups. However, we didn’t observe any trend in the direction of fold change with respect to milk-yield group. Next, in order to assess whether the overlap between the DEG and the SNP candidates are not due to random chance, we performed Hypergeometric tests on each category viz. low yield specific, high yield specific and common. Interestingly, in all categories p-value < 0.0001. Thus, considering that SNP candidate genes accounted more than half of the DEG in a cross-species comparison, we propose that buffalo SNP candidates potentially influence downstream gene expression changes to a large extend.

Figure 4 Comparison between DEG in high vs low-milk yield cattle and SNP candidate genes from buffalo breeds.

The top shows the workflow for the derivation of the DEG in high and low milk-yielding cattle using Agilent microarrays. The Venn diagram highlights the 582 DEG (55.1% of total) which are discovered SNP candidate genes discovered by our multi-breed study in buffaloes. The colored bar in indicates the group of SNP candidates in each category. Green color indicates 247 candidate genes common to high and low-yield specific groups, derived from specific SNPs in both the groups. Yellow and red color indicates the specific high-yield (139) and low-yield (196) candidate genes respectively. The overlap of the DEG and in each of these category were found to statistically significant in Hypergeometric test (p-value < 0.005).

Discussion

In India, milk or milk products form one of the primary source of nutrition in an average diet, and water buffalo’s milk contribute to about 55% country’s total milk production (Michelizzi et al., 2010). In this study, using Roche 454 GS-FLX Titanium sequencer we performed exome sequencing of replicated samples of three Indian buffalo breeds belonging to high or low milk-yield groups. The contrast between high and low-milk yield across breeds were about 3.7 fold, which was not influenced by disease, fertility or diet factors. The sequencing experiment generated about 2.49 Gb of data, and the reads were mapped to cattle genome, as the complete assembly of water buffalo whole genome sequence is not available yet. Initially, this resulted in high discovery rate of SNPs compared to the previously reports (Jansen et al., 2013; Georges, 2014). However, the number of SNPs have markedly reduced when we filtered out those which are specific for low and high milk-yielding group. It has been recently reported that there are indigenous bias for each SNP calling algorithm which causes disagreement of results across software tools (Pabinger et al., 2014; Yi et al., 2014). However, only few efforts have been taken to assess this aspect systematically (Yi et al., 2014). In this study, we have used three algorithms for SNP calling and have selected those SNP which were detected by at least two tools. Initially, in order to assess potential true positive calls, we have applied the widely accepted ts/tv metrics (DePristo et al., 2011). According to the 1000 genomes project, the expected ts/tv ratio in whole-genome sequencing is about 2.10, and for exome target regions ranges from 2.6 to 3.5 (http://www.1000genomes.org/). We have not only observed a lower ts/tv ratio in the non-reproducible SNP calls, but also a varying range of ts/tv values. In addition, the SNP calls from intergenic region were almost doubled in case of non-reproducible SNPs. Overall, we propose the MultiCom approach as a simple but effective strategy to identify reliable SNPs, without decreasing overall discovery rate.

After determining the SNPs specific for high and low milk-yield groups, it is important to understand its possible consequences by extending the analysis at gene level, which is a very useful strategy to obtain the possible molecular framework hosted by the SNPs (Lehne, Lewis & Schlitt, 2011; Menon & Farina, 2011; Shastry, 2009). Towards this, we focused on the genes in which SNPs fall in their exonic region. Initially, we have seen that majority of the milk-yield and milk-fat QTL candidate genes are found among the discovered group. The SNP candidate genes specifically found in the low-yield specific group includes ABCG2, CSN2, CSN3, IL12RB2, SRC, STAT5B etc. The classic milk proteins such as CSN2, CSN3 (members of casein family), STAT5B etc. are prominent indicators of milk yield parameters (Lee et al., 2014). Interestingly, in the low-yield specific group, there were several genes related to immunological properties of milk. For example, ABCG2 gene is responsible for the active secretion of clinically and toxicologically relevant substrates into the milk and soluble CD14 in colostrum and milk acts as a sentinel molecule and an immune modulator, which provide innate responses against bacterial infections in the calf (Olsen et al., 2007; Ibeagha-Awemu et al., 2008). IL12RB2 gene is another example in this category. Thus the low-yield specific group not only contained genes directly implicated in milk yield, but also comprised of genes that contribute to major immunological properties of milk. On the other hand, high-yield specific candidate genes included ARFGEF1, GH1, DGAT1, IRF9, PRL etc., which are hallmark genes for milk yield in cattle. Interestingly, Shi et al. (2012) have recently shown that the polymorphism of several of these genes affects the milk composition in different water buffalo breeds.

GO enrichment analysis showed several transcriptional regulation related terms in the low-milk yield group, and we highlighted the 140 genes responsible for this observation. Recently, a gene network analysis of human mammary transcriptome data highlighted crucial nodes such as NR1H3 and PPARA genes (Mohammad & Haymond, 2013). Notably, these genes are present in our results, and it was reported that PPARA and its target genes has been shown to be involved in fatty acid uptake/oxidation, and promotes energy balance during early lactation in cattle (Schlegel et al., 2012).

The pathway enrichment analysis pointed towards several known pathways implicated in mammary biology such as JAK-STAT signalling, Wnt signalling, epidermal growth factor related pathways (ErbB signalling, VEGF signalling) etc. (Watson & Burdon, 1996; Rossiter et al., 2007; Hardy et al., 2010; Turashvili et al., 2006). In the low milk-yield specific group, Oxidative phosphorylation pathway was the most enriched pathway, which may contribute to the energy balance during different functional states of mammary gland (Nelson, Butow & Ciaccio, 1962). Interestingly, there were two pathways common to high and low milk-yield groups. Considering that the input genes for the pathway analysis were different, Toll-like receptor signalling and MAP kinase signalling pathways were enriched by diverse set of genes belonging to the two groups. Very interestingly, O’Neill, Golenbock & Bowie, (2013) have extensively reviewed Toll-like receptor signalling pathway, and suggested the possible dual outcomes of this pathway. Two important families of transcription factors that are activated downstream of TLR signalling are nuclear factor-κB (NF-κB) and the interferon-regulatory factors (IRF3/7), and the major consequence of NFKB mediated signalling is the production of interleukins, while the IRFs induces type I interferon (O’Neill, Golenbock & Bowie, 2013). Notably, RELA (a member of NFKB complex) was present among low-specific candidates and IRF3 was present in the high milk-yield specific group. Of note, TLR-4 mediated excess interleukin production has shown to cause lactation insufficiency in mice (Glynn, Hutchinson & Ingman, 2014). On the other hand, increased interferon level was observed in breastfeeding infants compared to non-breast feeding group, which confers enhanced protection against viral infection for the infant (Melendi et al., 2010). However, more studies need to be performed in this direction, as evidenced by limited number of studies.

Comparison of gene expression and SNP candidate genes offers a great extent of biological insights (Quan et al., 2014). Recently, Cui et al. (2014) observed differential gene expression in various susceptibility genes related to milk protein and fat level in Holstein friesen cattle. In our case, the buffalo SNP data were identified based on cattle as the reference genome, and the gene expression data were from cattle. However, we excluded the putative generic SNPs in buffalo breeds (found common to both high and low-yield groups) at the initial stage of analysis which makes the comparison relevant, even though the genome and transcriptome data are from buffalo and cattle species respectively. In the comparison, we found a strong presence of high or low-yield specific SNP containing genes among the DEG. This observation not only underlines the strategy of comparison but also provides positive indication for the reliability of the identified SNPs which need to be investigated in additional datasets.

Conclusion

In this study, we have highlighted issues related to SNPs detection using a single tool and propose selection of reproducible SNPs using Multicom approach, which facilitates detection of reproducible SNPs with a higher discovery rate. Focussed analysis on the high and low milk-yield SNPs in buffalo breeds, uncovered SNPs specific for high or low milk yield. A substantial number of reported genes in QTL region of cattle genome is present in results. Further, the enrichment analysis uncovered several transcriptional regulator candidates in low-milk yield group. Surprisingly, pathway analysis shared pathways like TLR receptor signalling, along with several known and specific pathways in high and low-milk yield groups. Finally, SNP candidate genes in buffalo breeds account for were splendidly found in differentially expressed genes from the cattle species. On the whole, these results not only highlighted SNPs related to high and low-yield buffalo, but also sheds light to methodological improvement, functional insights and cross species genome-phenome comparison. Of course, outcomes of this study need to be validated in independent samples which in turn contribute for the better genomic selection in buffalo breeds.

Supplemental Information

Supplemental Information 1 Supplementary Table 1.

Click here for additional data file.

Supplemental Information 2 Supplementary Table 2.

Click here for additional data file.

Supplemental Information 3 Supplementary Table 3.

Click here for additional data file.

Supplemental Information 4 Supplementary Table 4.

Click here for additional data file.

Supplemental Information 5 Supplementary Fig. 1.

Click here for additional data file.

Supplemental Information 6 Supplementary Fig. 2.

Click here for additional data file.

Supplemental Information 7 Supplementary Fig. 3.

Click here for additional data file.

Additional Information and Declarations

Competing Interests

Author Contributions

Animal Ethics

Data Deposition

The authors declare that they have no competing interests.

Ramesh Menon conceived and designed the experiments, performed the experiments, analyzed the data, contributed reagents/materials/analysis tools, wrote the paper, prepared figures and/or tables, reviewed drafts of the paper.

Anand B. Patel performed the experiments, contributed reagents/materials/analysis tools.

Chaitanya Joshi conceived and designed the experiments, contributed reagents/materials/analysis tools, wrote the paper, reviewed drafts of the paper.

The following information was supplied relating to ethical approvals (i.e., approving body and any reference numbers):

Blood drawings from buffaloes were conducted in accordance with regulations and prior approval by the institutional animal ethics committee of Anand Agricultural University, Gujarat, India. It was conducted by an experienced veterinarian and the samples are freely available for academic research purpose.

The following information was supplied regarding data availability:

The data has been deposited in EBI-ENA repository: http://www.ebi.ac.uk/ena/data/view/PRJEB13801.

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
