# Peer review of "Comparative analysis of SNP candidates in disparate milk yielding river buffaloes using targeted sequencing"

_PeerJ, doi:10.7717/peerj.2147_

## Round 0.1 · original submission · Major Revisions

As you will notice, Reviewer 1 gives relatively positive comments while Reviewer 2 raises fundamental concerns on your experimental design and statistical analysis. Since I feel that Reviewer 2's opinion is reasonable, please read their comments carefully and revise the manuscript accordingly if you think that you will be able to satisfy both reviewers with your revised manuscript; otherwise, just submitting to another journal could be an option to be considered.

Reviewer 1 ·

Basic reporting

This study reports the detection of candidate genes for milk-yield with exome-sequencing using 12 water buffalo individuals. 12 animals of 3 breeds were composed with 2 high and 2 low milk-yield animals from each breed. Detection of SNPs was performed with 3 programs. Menon et al. selected SNPs shared with at least 2 programs, and showed that the method facilitates detection of reliable SNPs, not including false-positives.
Although, to detect SNPs, they used Bos taurus genome assembly as the reference sequence, detection of species and breeds specific SNPs was avoided by selecting high-milk yield specific or low-milk yield specific SNPs.
Furthermore, they considered that genes locating the high-yield specific or low-yield specific SNPs were candidate genes for milk-yield, and they showed that the candidate genes included many of known milk-yield and fat QTL candidate genes. They showed that the candidate genes might be related to milk-yield in water buffaloes by performing GO analysis and DEG analysis using transcriptome data for daily cattle obtained from database.
Overall, the manuscript is acceptable to publish. However, some points should be changed before publication.


Table2: Information for coverage is important in an analysis with next-generation sequencing, and the data shown in supplementary table 1 should be included in table 2.
It is preferable that data of “No. of Reads” were indicated like “total” data.

Figure 1: “High production” and “Low production” should be changed to “High yeild” and “Low yeild”.

Figure 3: Panel C and B are different from the figure legend. Panel C and B is miss-numbered.

L156: ”367,674 low yield specific 8284 coding genes” is ”the 367,674 low yield specific SNPs in 8284 coding genes”.

L159: ”the reported milk-yield milk-fat related genes in QTL region” is ”the reported milk-yield and milk-fat related genes in QTL region”.

L161: ”out of 74 milk-yield candidate genes, 30 were ” is ”of 74 milk-yield candidates, 51 gene were ”.

L163: ”out of 91 milk-fat candidates, 51 gene were ” is ”of 91 milk-fat candidates, 51 gene were ”.

Experimental design

No Comments

Validity of the findings

No Comments

Reviewer 2 ·

Basic reporting

1. The authors did not show detailed information about exome sequence statistics. Total Reads and coverage were shown in Table 2 and supplementary Table 1, but depth was not shown in these tables. In addition, coverage of sufficient depth to detect SNPs should be also indicated.

2. The authors did not explain about 12 samples used for genome analysis. How they select 12 buffalo? Is there any difference in age, gender, body weight, health status, and other factors between high and low milk yield cattle. I wonder if some genetic factors might affect these confounding factors. In addition, please explain about the methods how they measure total milk yield.

Minor comments
DEG was not explained in the manuscript.

Experimental design

3. The authors should explain about the definition of specific SNP. Specific for high yield groups means all 6 high yield buffalo carried this SNP, but no buffalo in low yield groups has this SNP? More the one million SNPs were identified by sequence analysis. The authors should evaluate the association of all SNPs with milk yield by using appropriate statistical method.

Validity of the findings

4. The result of sequence data should be validated by other methods. Please select one candidate SNPs and indicate the result of capillary sequence and its association with milk yield in 12 buffalo.

Additional comments

The authors conducted whole exome sequence of 12 water buffalo with high or low milk yield and identified more than one million SNP by using Roche 454 sequencer. In addition, they found nearly half million of high or low yield specific SNPs. Moreover, about 70% of milk-yield or milk-fat QTL were included in these candidate genes. These findings are potential interesting, but I have several concerns with the result and statistical methods using in this paper as shown in each box.

---

## Round 0.2 · Minor Revisions

Your revised manuscript is almost acceptable. Please read the suggestions from one of the reviewers and follow them, if you agree with them. Thanks for your patience.

Reviewer 1 ·

Basic reporting

There are minor comments.

In Table 2, please indicate the data by using “ , “, like “468,655”.

In Figure 1 and its legend, please change “high production” and “low production” to “high yield” and “low yield”, same as group name in the manuscript.

L90: Please also add a explanation for “SNO”.

L180: Please change “Figure 3B” to “Figure 3C”.

L218: Please change “2.49 gb” to “2.49 Gb”.

Experimental design

No Comments

Validity of the findings

No Comments

Additional comments

No Comments

Reviewer 2 ·

Basic reporting

No comments

Experimental design

No comments

Validity of the findings

No comments

Additional comments

The authors response to the comments of the reviewers. I considered that this paper is acceptable to PeerJ.

---

## Round 0.3 · accepted · Accept

Since you have followed all of the minor comments by one of the reviewers, I confirm that all points are addressed. Congratulations!